# MetaEmotions at School: A Program for Promoting Emotional and MetaEmotional Intelligence at School; a Research-Intervention Study

**Antonella D'Amico * and Alessandro Geraci**

Department of Psychology, Educational Science and Human Movement, University of Palermo, 90128 Palermo, Italy
* Correspondence: antonella.damico@unipa.it

**Abstract:** MetaEmotions at Schools is a SEL program that aims at promoting the culture of emotional and metaemotional intelligence in schools, and at improving emotional awareness both among teachers and students. It is a train-the-trainers program that unfolds in five steps, respectively, aimed at stimulating teachers and students to: (1) develop emotional literacy; (2) create emotionally inclusive environments; (3) build tools, materials and methods for emotionally inclusive classes; (4) develop didactic method mediated by emotions; (5) become ambassadors of the metaemotional intelligence at school and beyond. In this study, we report some results of the first application of the training program, focusing in particular on the effect of the program on emotional and metaemotional intelligence of participants. A total of 264 pupils from lower secondary schools of five Italian cities were recruited and divided in 9 intervention and 9 comparison classes. All participants were administered with emotional and metaemotional intelligence assessment tools before and after the training program. The results showed that emotional abilities scores in the intervention groups tended to be stable over time compared to the comparison groups ones which tended to decrease. Moreover, the pupils showed a reduced tendency to overestimate their emotional abilities. These results pave the way to further applications of the program and shed the light on areas of improvement.

**Keywords:** emotional intelligence; metaemotional intelligence; school; pupils; training; social-emotional learning; SEL program





## 1. Introduction

School is a microcosm in which children and adolescents interact with significant others outside the family environment thanks to whom they can improve their emotional knowledge and abilities and also experience their first social successes and failures. School must equally provide students with the opportunity to increase their cultural and academic knowledge, to improve their cognitive abilities, and to learn social and emotional skills proven to be necessary to cope with life [1–3].

This statement is shared by a large part of the scientific community and in the last few years there has been an increasing number of studies aimed at developing and applying methods for social emotional learning. A metanalysis study [4] analyzed as many as 200 interventions implemented in schools, and more studies have been conducted in recent years [5–10].

SEL programs are often based on different theoretical models, but share the general purpose of enabling children and adolescents to acquire social and emotional skills such as recognizing and using emotions, establishing and maintaining positive relationships, making responsible decisions, setting and achieving goals, being able to regulate their emotions and behavioral reactions and increasing and improving the quality of their social relationships [11,12]. The Collaborative for Academic, Social and Emotional Learning (CASEL), which is the leading organization promoting SEL programs in the education



systems, is the result of decades of collaboration between different scholars, namely, Daniel Goleman on the public side and Roger Weissberg on the scientific side [4,8]. Scholars adhering to CASEL [13] have also defined guidelines that the different SEL programs should follow, such as including socio-emotional education as a curricular teaching, providing differentiated teaching procedures, involving parents, involving and training teachers and school staff, demonstrating the quality of programs through research evidence.

Some SEL programs are based on the ability model by Mayer and Salovey [14], conceiving emotional intelligence as a set of cognitive abilities in perceiving, using, understanding and managing emotions. Among these programs the most known is the RULER (Recognizing, Understanding, Labeling, Expressing and Regulating emotion), which adheres to the CASEL guidelines, developed at the Center for Emotional Intelligence at Yale University by Marc Brackett and collaborators [15–18]. The RULER program includes several curricula aimed at recognizing, naming, expressing and regulating emotions. In Spain, thanks to the team led by Pablo Fernandez-Berrocal, the INTEMO program [19–21], also inspired by the Mayer-Salovey model [14], has been developed for some years now. In contrast with the RULER program, INTEMO is not only designed for school, but also for clinical or community contexts and must be applied by professionals competent in the field of emotional intelligence.

A further program inspired by the Mayer and Salovey model [14], is named MetaEmotions and has been developed in Italy by D'Amico and colleagues [22,23], starting from previous works and research [24,25] and it is the main focus of this study. MetaEmotions program is aligned with CASEL principles and it is organized in a slightly different way if compared to RULER and INTEMO. It presents a series of training sessions organized in five macro-areas, each one including one or more of the emotional skills described in Mayer and Salovey's model: (1) recognition and non-verbal expression of emotions; (2) emotional synesthesia; (3) understanding emotions; (4) emotional regulation; (5) use and genesis of emotions. MetaEmotions is currently designed in two versions: MetaEmotions Test & Training and MetaEmotions at School.

MetaEmotions Test & Training has been developed for applications in psychoeducational or psychotherapy contexts with small groups of children and adolescents; it aims to promote emotional abilities but, primarily, it aims at developing meta-emotional intelligence, intended as the awareness about one's own emotional abilities and the metaemotional beliefs [23]. People awareness about emotions and emotional abilities is an essential aspect of emotional life, since it affects the way that they put their emotional abilities into practice. Author [22,26] stated that often people present low metaemotional intelligence: they are not aware of their emotional intelligence and tend to overestimate or underestimate their emotional abilities. Moreover, they show to own a belief's system about emotions that is not consistent with scientific knowledge about emotions, not being, for instance, aware enough of the role of sensations or language for emotional life, or considering emotion as something that is not possible to manage. A recent study on pre-adolescents [27] found that the awareness of one's own emotional abilities is associated to sociometric status and that the meta-emotional beliefs are associated to personal well-being. In addition, another study [28] demonstrated that there are important sex differences in meta-emotional intelligence among pre-adolescents and adolescents, with girls tending to underestimate and boys to overestimate their emotional abilities.

For these reasons, the first phase of MetaEmotions Test & Training is the assessment of participants' emotional and metaemotional intelligence, performed using the IE-ACCME test [26], a multi-method multi-trait assessment tool that will be described in the method section. The assessment is followed by the discussion of results of each participant, that has the aim to stimulate metaemotional awareness on one's own strengths and possible areas for improvement. Thus, assessment is crucial in MetaEmotions Test & Training program: during each training session, participants are requested to reflect on results they achieved in the relative section of the IE-ACCME test in order to stimulate awareness and active strategies for self-improvement.

MetaEmotions at School, that we will focus on in this work, was instead developed for the school context and it aims to promote the culture of emotional and metaemotional intelligence among teachers and students.

For this reason, MetaEmotions at School uses a train-the-trainers methodology: teachers participate in training-experiential workshops to become ambassadors of emotional intelligence in their schools. They are then provided with a series of activities and guidelines, hosted in the web platform named MetaEmotions at School that they are invited to use, in order to conduct during school time activities of emotional literacy with their pupils and to use specific educational and teaching strategies.

By entering in the web platform, teachers become also part of the MetaEmotions community, helping to disseminate the program and to improve it. They are also requested to complete a board diary and to upload it in the platform at the end of activities with children, in order to share their experiences and suggestions with colleagues. Usually, MetaEmotions at School does not involve the assessment phase; however, in this study we measured emotional and meta-emotional intelligence of participants for research purposes.

*The Present Study*

The present study reports the results of the first application of the MetaEmotions at School program. In this study, we used a research-intervention method in order to examine the impact of MetaEmotions at School program used by lower secondary schools' teachers for promoting emotional and metaemotional intelligence of their students.

To this aim, teachers and students belonging to twelve Italian lower secondary schools were invited to participate to the study and were divided in 9 intervention and 9 comparison school classes. Next, the study unfolded in different steps. First, emotional and metaemotional intelligence were measured in students of intervention and comparison classes at the beginning of school year. Second, after the first testing session, teachers of intervention classes with a small representative of their students participated to training sessions with our experts, to learn how to use the MetaEmotions at School program. Third, MetaEmotions at School program was applied by teachers only in intervention classes for about three months, whereas the students in comparison classes performed normal school activities. Fourth, emotional and metaemotional intelligence were again measured in students of intervention and comparison classes at the end of school year.

Our hypothesis was to observe a higher improvement of emotional and metaemotional intelligence, intended as higher awareness of one's own emotional abilities, in students belonging to intervention classes, and lower improvement in comparison classes. Since different intervention classes did not follow exactly the same curriculum, analyses were performed both for the total group and for each intervention class separately.

## 2. Materials and Methods

A quasi-experimental two group pre-test and post-test research design was adopted to study the efficacy of MetaEmotions at School program in increasing emotional and meta emotional intelligence, in an intervention group of pupils following the program compared with a comparison group not involved in the activities.

*2.1. Participants*

A total of 264 pupils (M = 129, F = 135; average age = 11.95, SD = 0.27), belonging to second classes of twelve lower secondary schools from five Italian cities were involved in this study. Specifically, participants were from Turin (28.4%), Bari (21.6%), Cagliari (20.5%), Rome (17.4%), and Palermo (12.1%).

Participants were enrolled by contacting and inviting their schools to participate to the project, illustrating the whole program and the organization of the different phases of the project. For each school, we asked to Principals to choose two classes, assigning one on them to the intervention group and the second to the comparison group. Assignments were made by Principals also considering the availability and interest by teachers to participate

to the intervention experience. We decided to involve intervention and comparison classes belonging to the same schools in order to recruit participants form the same socio-cultural context and to reduce possible differences.

Principals of involved schools requested an informed consent to families of their students, with permission to take part in the study.

After selection and assignments, we had a total of 139 pupils (M = 74, F = 65; average age = 11.92, SD = 0.34) belonging to the 9 intervention classes, and a total of 125 pupils (M = 55, F = 70; average age = 11.99, SD = 0.16) belonging to the 9 comparison classes.

### 2.2. Procedure

The test phase took place in November–December 2017 and it was conducted by our experts in collective administration both in the comparison and intervention group. From January to May 2018 the intervention activities described below were carried out. The re-test phase took place in May 2018; the experts administered the same tests in collective administration both to intervention and comparison groups. The application of IE-ACCME test was not followed by an individual restitution of results (as it happens in MetaEmotions Test & Training method), since it was used only for research purposes.

### 2.3. Measures

Both in test and re-test session it was used the IE-ACCME Test [26] for measuring Emotional and Metaemotional Intelligence. The IE-ACCME Test is a multi-trait multi-method assessment tool, inspired by the four branches Mayer and Salovey [14] EI model, that allows to build a profile of emotional and metaemotional abilities in pre-adolescents and adolescents.

The instrument allows to measure the four branches of emotional intelligence (perception, facilitation, understanding and management of emotions) through different scales and measurement methods: the beliefs that an individual holds about emotions (CE self-report scale), the self-concept about one's own emotional skills (CME self-report scale), the actual emotional abilities (AE ability test) and the self-assessment of performance in the emotional ability test (AP self-report scale).

The Meta-Emotional Beliefs scale (CE) is a 16-item self-report scale with a 5-point Likert scale ranging from 0 ("not true") to 4 ("definitely true") that investigates individuals' beliefs about the role of emotions in everyday perceptions and sensations, in facilitating thought processes, as well as the possibility that emotions can be uniquely understood and regulated on both personal and interpersonal levels (e.g., "Complex feelings like love or friendship arise from a mixture of many emotions"). The Emotional Self-Concept scale (CME) is a 20-item self-report scale with 5-point Likert scale ranging from 0 ("not true") to 4 ("definitely true") that assesses one's perceived ability to recognize emotions in faces, images and sensations, to use emotions in thought processes, to understand the vocabulary and transformations of emotions, and to manage emotional states in the personal sphere and in relationships with others (e.g., "I am able to identify the emotions that derive from particular physical sensations"). The Emotional Ability Test (AE) is a maximum performance test that explores adolescents and pre-adolescents' performance in tasks of Perception of emotions, Facilitation of emotions in cognitive processes, Understanding of emotions and Management of emotional problem-solving situations. The test is similar but not overlapping to MSCEIT [29], it includes eight tasks grouped into the four branches of Mayer and Salovey model [14]. The scoring system is based on the consensus scoring method, and the consensus sample is composed of 1.084 Italian adolescents (M = 526, F = 558) between 10 and 19 years recruited from southern, central and northern Italy. The Self-Assessment of Performance scale (AP) is a 6-point Likert scale ranging from 0 ("not able") to 5 ("very able") that asks respondents to self-assess their own performance after each task foreseen in the Emotional Ability Test.

Validation and standardization of the IE-ACCME test was performed on the Italian consensus sample of 1.084 adolescents. Factorial analyses showed that all IE-ACCME scales

structures trace the Mayer and Salovey ability model [14]. Each scale presents acceptable reliabilities (test-retest: CE, r = 0.43, *p* < 0.001; CME, r = 0.76, *p* < 0.001; AE, r = 0.44, *p* < 0.001; AP, r = 0.55, *p* < 0.001; split-half: AE scale = 0.86). All IE-ACCME test scores are expressed as standardized scores with mean = 100 and standard deviation = 15.

A further important aspect of IE-ACCME test is that by combining standardized scores of CME, AE and AP scales, it is possible to compute two additional scores: metaemotional knowledge and metaemotional self-evaluation scores [27,28]. They are specifically aimed at exploring the awareness of preadolescents and adolescents about their emotional abilities in everyday and testing conditions. In particular, Metaemotional knowledge scores indicate how deeply people are aware of their own emotional abilities in everyday life and it is computed as the difference between standardized scores in Emotional Self-Concept scale (CME) and in Emotional Ability Test (AE), weighted by the scores in AE (final formula CMeta_Rel = CME − AE/AE). A negative value in Metaemotional knowledge scores indicate that respondents underestimate their emotional abilities in everyday life compared to the results obtained in the ability test, whereas a positive value indicate that they overestimate their emotional abilities in everyday life compared to the results obtained in the ability test.

Metaemotional self-evaluation scores indicate how deeply people are aware of their performance in the ability test and it is computed as the difference between standardized scores in Self-Assessment of Performance scale (AP) and in Emotional Ability Test (AE), weighted by the scores in AE (final formula AVMeta_Rel = AP − AE/AE). A negative value in Metaemotional self-evaluation scores indicate that respondents underestimated their performance in emotional abilities test, whereas a positive value indicate that they overestimated their performance.

### 2.4. The Training Program: MetaEmotions at School

As above mentioned, MetaEmotions at School is "train the trainers" program methodology aimed at help teachers to become facilitators of social emotional learning at school (SEL).

In this first study, we involved in the "train the trainers" activities two teachers and four students for each intervention class. The involvement of students was aimed at promoting a peer tutoring action in intervention classes. We recommended to teacher to choose the four students by random extraction, in order to avoid any preference between students.

The training program followed different steps that are briefly presented in Table 1 and described below.

**Table 1.** Brief description of MetaEmotions at School program.

| Step | Activity | Participants |
|---|---|---|
| I | Testing session I | Teachers and pupils from each intervention and comparison classes |
| II | Experiential workshop I | Representative teachers and pupils from each intervention classes |
| III | Class intervention | Teachers and pupils from each intervention classes |
| IV | Experiential workshop II | Representative teachers and pupils from each intervention classes |
| V | Testing session II | Teachers and pupils from each intervention and comparison classes |

*1. First experiential workshop.* After first testing session, selected teachers and pupils of each intervention class participated to an experiential workshop conducted by experts of our team. The first session of the experiential workshop lasted 8 h, during one whole day. The morning session (4 h) involved only teachers and was firstly aimed at explaining the theoretical background of MetaEmotions at School program and its organization. Teachers

were introduced to the theme of emotional and meta-emotional intelligence, focusing on the importance of such abilities for emotional well-being at school and for pupil's school achievement.

Finally, teachers were trained at using the online MetaEmotions at School platform, where they are provided with several activities to perform in their classes, along with materials and guidelines for conducting them.

The afternoon session also involved the four students selected for each intervention class, and was aimed at experience some activities drawn from MetaEmotions at School program, with the aim to train both teachers and student in conducting the activities and to encourage them to replicate them with classmates in the following days. During the workshop, teachers and pupils had the opportunity to access the heart of the program, being involved in many activities in order to familiarize with the program and to facilitate its application with the whole class.

Using a learning by doing methodology teachers and students were guided through the five goals of MetaEmotions at School program, such as: (1) discovering the world of emotions through emotional literacy; (2) creating emotional inclusive environment; (3) Building tools, materials and methods for emotionally inclusive schools; (4) Transforming the traditional teaching strategies into teaching strategies mediated by emotions; (5) Becoming ambassadors of emotional intelligence.

To achieve the first goal several activities about recognition and non-verbal expression of emotions, emotional synesthesia, understanding of emotions, management of emotions, were performed during classroom lessons. To create emotional inclusive environments teachers and students were guided to reflect on some basic aspect of environmental psychology, and on the role that the physical context (e.g., the classroom, the desk disposition) exert on their well-being and learning; next, they were instructed in a series of strategies for improving their own school environment. Teachers and students have then been guided and invited to use and to build tools, materials and methods for emotionally inclusive schools with their students. To this aim, they have been presented with the emotional diary, the thermometer of emotions to be displayed in the classroom, a selection of readings, stories, or videos for the creation of a themed film club. Teachers and students have been guided on the choice and use of specific methodologies allowing to transform the traditional teaching strategies into teaching strategies mediated by emotions. This step is particularly important because it allows teachers and pupils to understand that emotions are not a simple corollary to "feeling good" while learning, but are themselves instruments of learning, and everything that is conveyed by emotions, in fact, is learned with greater ease and durability. Some examples are the use of storytelling, dramatization, or the use of songs for learnings. The final goal of the program has been then to urge teacher and students to become ambassadors of emotional intelligence also outside the school by involving friends and family in some of the activities, for instance by collaborating to realize tools or by sharing with parents teaching strategies mediated by emotions for the homework, or to organize events in order "to export" the culture of emotional intelligence outside the school. For this reason, at the end of the workshop, students were named "ambassadors of emotional intelligence" and were invested as trainers for their classmate, in order to activate a peer-to-peer tutoring.

*2. Class activities.* For about three months after the first training, intervention classes were involved in a series of activities ì guided by teachers and by the "ambassadors of emotional intelligence". Each class was assigned with an expert tutor from our team, that coached teachers in implementing the activities. Many activities on emotional literacy were drawn as-they-were from the MetaEmotions at school platform. Other activities were adapted by teachers or were created applying our guidelines to specific disciplines. Thus, for instance, teachers of Arts used the activities suggested in the area of emotional synesthesia to realize drawing or painting activities with students; other teachers used our guideline on teaching strategies mediated by emotions, using songs or storytelling for supporting learning of poetry or history; a class had the opportunity to improve emotional

inclusive environment in their classroom by painting the walls of different color that represented the students' moods; other classes started to play instrumental music during math activities or other group activities, in order to create an emotional climate favorable for learning. During the whole period, teachers were asked to write an on-board diary in which they described all the activities they performed—with a final balance of the experience—and to upload it in the MetaEmotions at school platform. As expected, the intervention classes did not follow the exact same curriculum. Teachers had the opportunity to choose among a very rich number of activities in MetaEmotions platform, and they were also encouraged to adapt and create new activities depending on the specific needs of their students.

*3. Second experiential workshop.* At the end of the activities teachers and students attended to a focus group with experts of our team. The focus group lasted 8 h and was performed in a whole day, but was again divided in two sessions. The morning session was attended only by teachers, while the students participated to the afternoon session.

During the morning session, we discussed with teacher the undergone activities, the difficulties they eventually have had, and the feedbacks obtained by students and their families; teachers were also asked to complete an open-ended questionnaire for evaluating the entire project.

During the afternoon session students were interviewed by experts in our team, asking them to report the most significant situation they experienced during the project, how the activities had led them to reflect about emotions and also to report some critical aspect of the experience. Then they were as well asked to complete an open-ended questionnaire for evaluating the entire project.

At the end of the workshop teacher and students created also together a poster with drawings and messages related to the project but also photos, images or cards that were made during the activities carried out during the school year with the teachers.

## 3. Results

In order to highlight the expected differences between the intervention group (IG) and the comparison group (CG) in all measured variables before and after the application of the MetaEmotions at School program, statistical data analyses were carried out on the scores obtained in the tests administered to pupils during the test and retest phases.

The first series of analyses were performed in the total sample (9 intervention and 9 comparison classes) and had the aim of assessing whether and to what extent all students belonging to the intervention classes had reported, in the retest phase, an improvement in all the studied variables greater than those observed in all students belonging to the CG.

Since, as above mentioned, the intervention was performed in different ways in each intervention class (IC) depending on the choices of teachers involved and on the needs of students, we also performed a series of separate repeated measures ANOVAs for each intervention class. Results of each EC were compared to the total CG ones (Table 2): a general CG was considered more representative of population involved in the research than 10 separate CG that were often composed of a different number of students.

**Table 2.** Repeated measures ANOVA results for the overall sample and each intervention class.

| | Intervention Group (n = 139) | | | | Comparison Group (n = 125) | | | | ANOVAs | | | | | | | | | | | |
| | Test | | Retest | | Test | | Retest | | Group | | | | Time | | | | Time × Group | | | |
| **Scale** | M | SD | M | SD | M | SD | M | SD | $F_{(1, 262)}$ | $p$ | $\eta_p^2$ | d | $F_{(1, 262)}$ | $p$ | $\eta_p^2$ | d | $F_{(1, 262)}$ | $p$ | $\eta_p^2$ | d |
|---|---|---|---|---|---|---|---|---|---|---|---|---|---|---|---|---|---|---|---|---|
| **Total Group** | | | | | | | | | | | | | | | | | | | | |
| CE | 96.08 | 15.42 | 96.58 | 15.16 | 102.15 | 15.25 | 98.61 | 15.81 | 7.06 | 0.008 | 0.03 | 0.26 | 1.80 | 0.181 | 0.01 | 0.09 | 3.16 | 0.076 | 0.01 | 0.13 |
| CME | 97.09 | 16.49 | 98.52 | 17.70 | 102.79 | 15.62 | 101.96 | 16.38 | 6.99 | 0.009 | 0.03 | 0.28 | 0.08 | 0.784 | 0.00 | 0.00 | 1.06 | 0.304 | 0.00 | 0.06 |
| AE | 95.14 | 15.99 | 94.49 | 17.21 | 101.43 | 17.63 | 95.30 | 16.55 | 4.35 | 0.038 | 0.02 | 0.29 | 8.09 | 0.005 | 0.03 | 0.20 | 5.31 | 0.022 | 0.02 | 0.16 |
| AP | 104.17 | 14.11 | 84.87 | 12.69 | 103.65 | 13.39 | 87.17 | 10.67 | 0.45 | 0.504 | 0.00 | 0.06 | 448.12 | 0.000 | 0.63 | 1.40 | 2.79 | 0.096 | 0.01 | 0.09 |
| CMeta_Rel | 0.04 | 0.22 | 0.07 | 0.25 | 0.04 | 0.23 | 0.10 | 0.25 | 0.30 | 0.587 | 0.00 | 0.06 | 8.15 | 0.005 | 0.03 | 0.18 | 0.86 | 0.354 | 0.00 | 0.06 |
| AVMeta_Rel | 0.13 | 0.25 | −0.07 | 0.21 | 0.05 | 0.20 | −0.06 | 0.18 | 2.21 | 0.138 | 0.01 | 0.14 | 118.65 | 0.000 | 0.31 | 0.72 | 10.32 | 0.001 | 0.04 | 0.20 |
| **Intervention Class 1 (n = 16)** | | | | | | | | | $F_{(1, 139)}$ | $p$ | $\eta_p^2$ | d | $F_{(1, 139)}$ | $p$ | $\eta_p^2$ | d | $F_{(1, 139)}$ | $p$ | $\eta_p^2$ | d |
| CE | 99.03 | 18.59 | 93.90 | 15.75 | | | | | 1.38 | 0.243 | 0.01 | 0.16 | 2.97 | 0.087 | 0.02 | 0.18 | 0.10 | 0.751 | 0.00 | 0.00 |
| CME | 90.51 | 19.27 | 90.56 | 16.06 | | | | | 10.11 | 0.002 | 0.07 | 0.47 | 0.03 | 0.856 | 0.00 | 0.00 | 0.04 | 0.840 | 0.00 | 0.00 |
| AE | 105.75 | 21.90 | 91.49 | 27.79 | | | | | 0.00 | 0.948 | 0.00 | 0.00 | 13.74 | 0.000 | 0.09 | 0.36 | 2.18 | 0.142 | 0.02 | 0.14 |
| AP | 105.74 | 15.88 | 83.13 | 14.79 | | | | | 0.12 | 0.726 | 0.00 | 0.00 | 117.61 | 0.000 | 0.46 | 1.00 | 2.89 | 0.091 | 0.02 | 0.14 |
| CMeta_Rel | −0.11 | 0.24 | 0.08 | 0.37 | | | | | 2.23 | 0.138 | 0.02 | 0.22 | 16.36 | 0.000 | 0.11 | 0.31 | 4.84 | 0.030 | 0.03 | 0.17 |
| AVMeta_Rel | 0.04 | 0.27 | −0.03 | 0.30 | | | | | 0.11 | 0.739 | 0.00 | 0.06 | 8.55 | 0.004 | 0.06 | 0.28 | 0.46 | 0.499 | 0.00 | 0.06 |
| **Intervention Class 2 (n = 10)** | | | | | | | | | $F_{(1, 133)}$ | $p$ | $\eta_p^2$ | d | $F_{(1, 133)}$ | $p$ | $\eta_p^2$ | d | $F_{(1, 133)}$ | $p$ | $\eta_p^2$ | d |
| CE | 97.34 | 21.68 | 95.32 | 21.76 | | | | | 0.88 | 0.350 | 0.01 | 0.13 | 0.84 | 0.361 | 0.01 | 0.09 | 0.06 | 0.803 | 0.00 | 0.00 |
| CME | 105.48 | 12.22 | 111.12 | 14.52 | | | | | 1.73 | 0.190 | 0.01 | 0.20 | 0.84 | 0.361 | 0.01 | 0.09 | 1.52 | 0.220 | 0.01 | 0.11 |
| AE | 106.54 | 20.86 | 104.79 | 21.56 | | | | | 2.39 | 0.125 | 0.02 | 0.22 | 1.50 | 0.224 | 0.01 | 0.13 | 0.46 | 0.497 | 0.00 | 0.06 |
| AP | 93.93 | 12.66 | 83.96 | 13.51 | | | | | 3.84 | 0.052 | 0.03 | 0.27 | 34.22 | 0.000 | 0.21 | 0.57 | 2.07 | 0.152 | 0.02 | 0.13 |
| CMeta_Rel | 0.01 | 0.17 | 0.09 | 0.19 | | | | | 0.09 | 0.765 | 0.00 | 0.06 | 2.87 | 0.093 | 0.02 | 0.14 | 0.04 | 0.843 | 0.00 | 0.00 |
| AVMeta_Rel | −0.09 | 0.20 | −0.18 | 0.15 | | | | | 5.98 | 0.016 | 0.04 | 0.35 | 7.86 | 0.006 | 0.06 | 0.27 | 0.05 | 0.824 | 0.00 | 0.00 |
| **Intervention Class 3 (n = 15)** | | | | | | | | | $F_{(1, 138)}$ | $p$ | $\eta_p^2$ | d | $F_{(1, 138)}$ | $p$ | $\eta_p^2$ | d | $F_{(1, 138)}$ | $p$ | $\eta_p^2$ | d |
| CE | 94.58 | 13.18 | 94.18 | 15.91 | | | | | 3.30 | 0.072 | 0.02 | 0.24 | 0.56 | 0.454 | 0.00 | 0.09 | 0.35 | 0.553 | 0.00 | 0.06 |
| CME | 105.09 | 12.68 | 98.32 | 15.44 | | | | | 0.03 | 0.855 | 0.00 | 0.00 | 2.81 | 0.096 | 0.02 | 0.16 | 1.72 | 0.192 | 0.01 | 0.11 |
| AE | 90.78 | 6.81 | 88.99 | 18.34 | | | | | 5.23 | 0.024 | 0.04 | 0.31 | 2.15 | 0.145 | 0.02 | 0.14 | 0.65 | 0.423 | 0.01 | 0.09 |
| AP | 107.19 | 8.98 | 88.92 | 14.87 | | | | | 0.96 | 0.329 | 0.01 | 0.13 | 81.18 | 0.000 | 0.37 | 0.89 | 0.22 | 0.644 | 0.00 | 0.00 |
| CMeta_Rel | 0.16 | 0.14 | 0.12 | 0.18 | | | | | 1.64 | 0.203 | 0.01 | 0.19 | 0.12 | 0.732 | 0.00 | 0.00 | 2.25 | 0.136 | 0.02 | 0.13 |
| AVMeta_Rel | 0.19 | 0.12 | 0.03 | 0.25 | | | | | 7.34 | 0.008 | 0.05 | 0.36 | 17.43 | 0.000 | 0.11 | 0.41 | 0.45 | 0.503 | 0.00 | 0.06 |

**Table 2.** *Cont.*

| | | | | | | **Total Group** | | | | | | | | | | |
|---|---|---|---|---|---|---|---|---|---|---|---|---|---|---|---|---|
| **Intervention Class 4 (n = 12)** | | | | | | $F_{(1, 135)}$ | *p* | $\eta_p^2$ | **d** | $F_{(1, 135)}$ | *p* | $\eta_p^2$ | **d** | $F_{(1, 135)}$ | *p* | $\eta_p^2$ | **d** |
| CE | 91.99 | 14.55 | 96.92 | 15.86 | | 2.54 | 0.113 | 0.02 | 0.22 | 0.06 | 0.807 | 0.00 | 0.00 | 2.19 | 0.141 | 0.02 | 0.16 |
| CME | 87.19 | 11.57 | 89.92 | 18.35 | | 11.55 | 0.001 | 0.08 | 0.49 | 0.14 | 0.714 | 0.00 | 0.00 | 0.47 | 0.493 | 0.00 | 0.06 |
| AE | 79.36 | 4.58 | 92.05 | 9.89 | | 9.61 | 0.002 | 0.07 | 0.43 | 1.29 | 0.258 | 0.01 | 0.11 | 10.66 | 0.001 | 0.07 | 0.31 |
| AP | 111.77 | 14.94 | 85.12 | 11.52 | | 0.98 | 0.323 | 0.01 | 0.13 | 108.35 | 0.000 | 0.45 | 1.00 | 6.03 | 0.015 | 0.04 | 0.21 |
| CMeta_Rel | 0.10 | 0.16 | 0.00 | 0.28 | | 0.10 | 0.751 | 0.00 | 0.06 | 0.37 | 0.545 | 0.00 | 0.06 | 4.52 | 0.035 | 0.03 | 0.19 |
| AVMeta_Rel | 0.41 | 0.17 | −0.06 | 0.20 | | 13.97 | 0.000 | 0.09 | 0.47 | 76.85 | 0.000 | 0.36 | 0.80 | 29.53 | 0.000 | 0.18 | 0.47 |
| **Intervention Class 5 (n = 12)** | | | | | | $F_{(1, 135)}$ | *p* | $\eta_p^2$ | **d** | $F_{(1, 135)}$ | *p* | $\eta_p^2$ | **d** | $F_{(1, 135)}$ | *p* | $\eta_p^2$ | **d** |
| CE | 92.63 | 16.37 | 99.54 | 11.15 | | 1.33 | 0.250 | 0.01 | 0.16 | 0.36 | 0.549 | 0.00 | 0.06 | 3.45 | 0.065 | 0.03 | 0.19 |
| CME | 90.19 | 16.60 | 101.91 | 17.36 | | 2.39 | 0.125 | 0.02 | 0.22 | 4.34 | 0.039 | 0.03 | 0.19 | 5.76 | 0.018 | 0.04 | 0.22 |
| AE | 90.02 | 8.82 | 99.62 | 19.84 | | 0.71 | 0.402 | 0.01 | 0.11 | 0.35 | 0.553 | 0.00 | 0.06 | 7.28 | 0.008 | 0.05 | 0.26 |
| AP | 108.84 | 17.56 | 87.37 | 13.53 | | 0.72 | 0.398 | 0.01 | 0.11 | 88.30 | 0.000 | 0.40 | 0.87 | 1.52 | 0.219 | 0.01 | 0.11 |
| CMeta_Rel | 0.01 | 0.20 | 0.06 | 0.26 | | 0.31 | 0.578 | 0.00 | 0.09 | 2.24 | 0.137 | 0.02 | 0.13 | 0.01 | 0.934 | 0.00 | 0.00 |
| AVMeta_Rel | 0.22 | 0.24 | −0.08 | 0.27 | | 2.14 | 0.145 | 0.02 | 0.20 | 40.21 | 0.000 | 0.23 | 0.58 | 8.92 | 0.003 | 0.06 | 0.26 |
| **Intervention Class 6 (n = 20)** | | | | | | $F_{(1, 143)}$ | *p* | $\eta_p^2$ | **d** | $F_{(1, 143)}$ | *p* | $\eta_p^2$ | **d** | $F_{(1, 143)}$ | *p* | $\eta_p^2$ | **d** |
| CE | 100.10 | 16.82 | 97.67 | 18.06 | | 0.23 | 0.630 | 0.00 | 0.06 | 1.80 | 0.182 | 0.01 | 0.13 | 0.06 | 0.803 | 0.00 | 0.00 |
| CME | 104.29 | 16.87 | 98.15 | 21.31 | | 0.12 | 0.734 | 0.00 | 0.06 | 2.97 | 0.087 | 0.02 | 0.14 | 1.73 | 0.190 | 0.01 | 0.11 |
| AE | 95.84 | 15.01 | 95.61 | 12.28 | | 0.63 | 0.427 | 0.00 | 0.11 | 1.97 | 0.163 | 0.01 | 0.13 | 1.70 | 0.195 | 0.01 | 0.13 |
| AP | 107.36 | 13.34 | 84.96 | 13.15 | | 0.09 | 0.762 | 0.00 | 0.00 | 140.84 | 0.000 | 0.50 | 1.10 | 3.26 | 0.073 | 0.02 | 0.14 |
| CMeta_Rel | 0.11 | 0.23 | 0.05 | 0.28 | | 0.01 | 0.907 | 0.00 | 0.00 | 0.01 | 0.920 | 0.00 | 0.00 | 4.42 | 0.037 | 0.03 | 0.17 |
| AVMeta_Rel | 0.15 | 0.25 | −0.10 | 0.18 | | 0.70 | 0.405 | 0.01 | 0.11 | 44.82 | 0.000 | 0.24 | 0.62 | 6.75 | 0.010 | 0.05 | 0.23 |
| **Intervention Class 7 (n = 16)** | | | | | | $F_{(1, 139)}$ | *p* | $\eta_p^2$ | **d** | $F_{(1, 139)}$ | *p* | $\eta_p^2$ | **d** | $F_{(1, 139)}$ | *p* | $\eta_p^2$ | **d** |
| CE | 98.24 | 11.66 | 104.32 | 12.43 | | 0.08 | 0.782 | 0.00 | 0.00 | 0.28 | 0.597 | 0.00 | 0.06 | 4.00 | 0.048 | 0.03 | 0.20 |
| CME | 96.08 | 13.26 | 108.12 | 13.21 | | 0.01 | 0.939 | 0.00 | 0.00 | 7.03 | 0.009 | 0.05 | 0.23 | 9.25 | 0.003 | 0.06 | 0.26 |
| AE | 100.89 | 13.90 | 96.03 | 11.16 | | 0.00 | 0.978 | 0.00 | 0.00 | 4.66 | 0.033 | 0.03 | 0.21 | 0.06 | 0.802 | 0.00 | 0.00 |
| AP | 96.52 | 11.75 | 82.08 | 13.57 | | 5.14 | 0.025 | 0.04 | 0.30 | 75.39 | 0.000 | 0.35 | 0.80 | 0.33 | 0.568 | 0.00 | 0.06 |
| CMeta_Rel | −0.02 | 0.22 | 0.14 | 0.17 | | 0.07 | 0.800 | 0.00 | 0.00 | 12.98 | 0.000 | 0.09 | 0.29 | 2.97 | 0.087 | 0.02 | 0.14 |
| AVMeta_Rel | −0.02 | 0.23 | −0.13 | 0.19 | | 2.49 | 0.117 | 0.02 | 0.22 | 14.62 | 0.000 | 0.10 | 0.36 | 0.00 | 0.964 | 0.00 | 0.00 |

**Table 2.** *Cont.*

| | | | | | Total Group | | | | | | | | | | | |
|---|---|---|---|---|---|---|---|---|---|---|---|---|---|---|---|---|
| **Intervention Class 8 (n = 20)** | | | | | $F_{(1, 143)}$ | *p* | $\eta_p^2$ | d | $F_{(1, 143)}$ | *p* | $\eta_p^2$ | d | $F_{(1, 143)}$ | *p* | $\eta_p^2$ | d |
| CE | 94.45 | 13.09 | 94.82 | 13.35 | 3.75 | 0.055 | 0.03 | 0.26 | 0.53 | 0.466 | 0.00 | 0.06 | 0.81 | 0.370 | 0.01 | 0.09 |
| CME | 94.89 | 18.54 | 93.21 | 17.53 | 5.94 | 0.016 | 0.04 | 0.36 | 0.42 | 0.516 | 0.00 | 0.06 | 0.05 | 0.825 | 0.00 | 0.00 |
| AE | 94.65 | 13.11 | 94.74 | 14.30 | 1.25 | 0.266 | 0.01 | 0.16 | 1.70 | 0.195 | 0.01 | 0.13 | 1.81 | 0.181 | 0.01 | 0.13 |
| AP | 101.16 | 13.50 | 84.54 | 9.88 | 1.12 | 0.292 | 0.01 | 0.13 | 106.50 | 0.000 | 0.43 | 0.95 | 0.00 | 0.964 | 0.00 | 0.00 |
| CMeta_Rel | 0.02 | 0.24 | 0.00 | 0.23 | 1.24 | 0.267 | 0.01 | 0.17 | 0.49 | 0.485 | 0.00 | 0.06 | 1.97 | 0.162 | 0.01 | 0.11 |
| AVMeta_Rel | 0.08 | 0.18 | −0.09 | 0.18 | 0.02 | 0.880 | 0.00 | 0.00 | 28.74 | 0.000 | 0.17 | 0.50 | 1.33 | 0.251 | 0.01 | 0.11 |
| **Intervention Class 9 (n = 18)** | | | | | $F_{(1, 141)}$ | *p* | $\eta_p^2$ | d | $F_{(1, 141)}$ | *p* | $\eta_p^2$ | d | $F_{(1, 141)}$ | *p* | $\eta_p^2$ | d |
| CE | 94.47 | 15.29 | 93.29 | 12.32 | 4.37 | 0.038 | 0.03 | 0.29 | 1.04 | 0.309 | 0.01 | 0.11 | 0.26 | 0.610 | 0.00 | 0.06 |
| CME | 98.17 | 15.38 | 99.99 | 15.96 | 0.90 | 0.345 | 0.01 | 0.14 | 0.06 | 0.806 | 0.00 | 0.00 | 0.43 | 0.514 | 0.00 | 0.06 |
| AE | 91.57 | 16.33 | 91.36 | 15.47 | 3.86 | 0.052 | 0.03 | 0.27 | 1.69 | 0.195 | 0.01 | 0.13 | 1.48 | 0.226 | 0.01 | 0.11 |
| AP | 104.39 | 12.94 | 84.47 | 11.70 | 0.15 | 0.702 | 0.00 | 0.06 | 121.00 | 0.000 | 0.46 | 1.00 | 1.08 | 0.300 | 0.01 | 0.09 |
| CMeta_Rel | 0.09 | 0.18 | 0.11 | 0.21 | 0.35 | 0.554 | 0.00 | 0.09 | 1.98 | 0.162 | 0.01 | 0.11 | 0.32 | 0.572 | 0.00 | 0.06 |
| AVMeta_Rel | 0.17 | 0.25 | −0.06 | 0.14 | 2.44 | 0.120 | 0.02 | 0.20 | 37.24 | 0.000 | 0.21 | 0.59 | 5.00 | 0.027 | 0.03 | 0.21 |

*Note.* CE = Metaemotional Beliefs; CME = Self-Perceived EI; AE = Emotional Intelligence; AP = Self-Evaluation in the EI Test; CMeta_Rel = Metaemotional Knowledge; AVMeta_Rel = Metaemotional Self-Evaluation.

All repeated measures ANOVAs (the one involving only the total samples and the others involving separately each intervention class) had two levels of the variable between Groups (Intervention/Comparison) and two levels of the variable Time (Test/Retest). The entered variables were the total scores of the measured variables, meta-emotional beliefs (CE), self-perceived EI (CME), emotional abilities (AE), self-rating about performance (AP), meta-emotional knowledge (CMeta_Rel), and meta-emotional self-evaluation (AVMeta_Rel).

### 3.1. Total Sample Results

The results of analyses performed on the total sample are reported in Table 2, and showed a significant small Group effect on CE [$F_{(1, 262)}$ = 7.06, $p < 0.01$, $\eta_p^2$ = 0.03, $d$ = 0.26], but no significant Time effect nor interaction effect Time × Group was found. Same results were found for CME showing only a significant small Group effect [$F_{(1, 262)}$ = 6.99, $p < 0.05$, $\eta_p^2$ = 0.03, $d$ = 0.28]. These indicate that participants in the IG presented lower mean scores for CE and CME than those in the CG. For emotional intelligence abilities (AE) a significant small effect Group effect [$F_{(1, 262)}$ = 4.35, $p < 0.05$, $\eta_p^2$ = 0.02, $d$ = 0.29] and a small significant Time effect [$F_{(1, 262)}$ = 8.09, $p < 0.01$, $\eta_p^2$ = 0.03, $d$ = 0.20] were found indicating that compared to the CG, IG mean scores for AE were lower. In addition, a small significant interaction Time × Group effect was found [$F_{(1, 262)}$ = 5.31, $p < 0.05$, $\eta_p^2$ = 0.02, $d$ = 0.16] indicating that emotional intelligence abilities scores decreased over time in both groups, particularly for the CG. Moreover, for the self-evaluation of performance in the ability test (AP), results showed a large significant Time effect [$F_{(1, 262)}$ = 448.12, $p < 0.001$, $\eta_p^2$ = 0.63, $d$ = 1.40], meaning that in between testing situations the self-evaluations decreased.

Concerning metaemotional abilities, the results for CMeta_Rel showed only a small significant Time effect [$F_{(1, 262)}$ = 8.15, $p < 0.01$, $\eta_p^2$ = 0.03, $d$ = 0.18], indicating that participants increased slightly their levels of overestimation over time. No group differences were found.

Results for AVMeta_Rel showed a large Time effect [$F_{(1, 262)}$ = 118.65, $p < 0.001$, $\eta_p^2$ = 0.31, $d$ = 0.72] and a small interaction Time × Group effect [$F_{(1, 262)}$ = 10.32, $p < 0.01$, $\eta_p^2$ = 0.04, $d$ = 0.20] indicating that over time the IG participants have shifted from the overestimation to underestimation of emotional abilities more than the CG.

### 3.2. Separate Intervention Classes Results

Table 2 presents also descriptive statistics of the total scores of all the variables for each of the IC (from IC1 to IC9) participating to the study, and results of the 2 × 2 repeated measure ANOVAs. Focusing exclusively on the Time × Group interaction effect for each of the IC, in IC1 a small interaction effect for CMeta_Rel [$F_{(1, 139)}$ = 4.84, $p < 0.01$, $\eta_p^2$ = 0.03, $d$ = 0.17] was found, indicating that over time intervention group has shifted from underestimation to overestimation more than the CG. On the contrary, in between testing sessions, IC4 pupils showed an increase in AE [$F_{(1, 135)}$ = 10.66, $p < 0.01$, $\eta_p^2$ = 0.07, $d$ = 0.31], a decrease in AP [$F_{(1, 135)}$ = 6.03, $p < 0.05$, $\eta_p^2$ = 0.04, $d$ = 0.21], and a decrease in CMeta_Rel [$F_{(1, 135)}$ = 4.52, $p < 0.05$, $\eta_p^2$ = 0.03, $d$ = 0.19] and in AVMeta_Rel [$F_{(1, 135)}$ = 29.53, $p < 0.001$, $\eta_p^2$ = 0.18, $d$ = 0.47]. These results indicate that IC4 pupils, over time, have increased their emotional abilities and the also have reduced their tendency to overestimate their emotional abilities both in everyday life and in testing situation. Results for IC5 showed an increase in CME [$F_{(1, 135)}$ = 5.76, $p < 0.05$, $\eta_p^2$ = 0.04, $d$ = 0.22] and in AE scores [$F_{(1, 135)}$ = 7.28, $p < 0.01$, $\eta_p^2$ = 0.05, $d$ = 0.26], and a decrease in AVMeta_Rel [$F_{(1, 135)}$ = 8.92, $p < 0.01$, $\eta_p^2$ = 0.06, $d$ = 0.26]. Thus, IC5 pupils increased both their perceived and actual emotional abilities, and furthermore, they reduced their tendency to overestimate their emotional abilities in testing situation. For IC6 pupils, results showed a slight decrease CMeta_Rel [$F_{(1, 143)}$ = 4.42, $p < 0.05$, $\eta_p^2$ = 0.03, $d$ = 0.17] and AVMeta_Rel [$F_{(1, 143)}$ = 6.75, $p < 0.05$, $\eta_p^2$ = 0.05, $d$ = 0.23], indicating that over time pupils have reduced their tendency to overestimate their emotional abilities in everyday life situations and in testing situations. IC7 pupils showed an increase in self-perceived EI [CME: $F_{(1, 139)}$ = 9.25, $p < 0.01$, $\eta_p^2$ = 0.06,

$d = 0.22$] whereas IC9 showed a decrease in AVMeta_Rel score [AVMeta_Rel: $F_{(1, 141)} = 5.00$, $p < 0.05$, $\eta_p^2 = 0.03$, $d = 0.21$], indicating that they reduced their tendency to overestimate their emotional abilities in testing situations

## 4. Discussion

Results of analyses performed on the total sample showed that students in IG presented different results than students in CG for emotional abilities scores, and in both metaemotional knowledge and metaemotional self-evaluation. In general, these results indicate a surprising effect. They seem to suggest that emotional abilities in total IG are more stable over time. Probably, these results indicate that IG students were more attentive, accurate and motivated in completing the test after the training than students in CG.

Over time, all students became more aware of their emotional abilities in performing the emotional ability test. Both students in IG and CG reduced their tendency to overestimate their performance in the ability test, shifting towards underestimation in meta-emotional self-evaluation. More likely that indicates that, in both groups there was a higher awareness in retest session about the difficulties of the test (i.e., the emotional ability score decreased form test to retest). However, the Group x Time interaction indicates that this effect, was more evident in IG than in CG.

Concerning results obtained in each IC, results evidenced the expected variation between them. From test to retest session, we obtained some significant difference between IC and CG only for IC1, IC4, IC5, IC6 IC7 and IC9, while no significant difference in emotional or metaemotional variables were found between IC2, IC3 and IC8 and CG in test and retest sessions. Consequently, analyzing results of each group separately was more informative for giving a picture of the different effect obtained by the MetaEmotions at School program in different classes and contexts. Moreover, variation in IC1 and IC7 were limited to one variable and do not seem particularly interesting.

On the contrary, we observed a very important improvement in emotional abilities for IC4 and IC5; moreover, they had important results in metaemotional variables since students' tendency to overestimate their emotional abilities in testing condition showed in test session disappeared in retest session. IC6 did not show an improvement in emotional ability but resulted more precautionary in their emotional self-concepts. Thus, they shifted from overestimation to underestimation of their emotional abilities both in everyday and in testing situations, demonstrating a more attentive approach to own emotions. Additionally, in IC9, finally, both students in IG and CG reduced their tendency to overestimate their performance in the ability test form test to retest, and the Group $\times$ Time interaction indicates that this effect was more evident in IG than in CG.

*Limitations*

A limitation of this study is the short duration of the program. Activities for students lasted for about three months but it is well acknowledged that emotional growth is a lifelong learning process which requires to be a part of school curricula from early childhood to early adulthood. Another limitation concerns the fact that, as in all research-intervention carried out at school, the intervention and comparison groups can never be said to be balanced nor random, as they are in fact made up of the classes, which always have a very different history and different achievements. It may depend on the origin of the students, on the relational dynamics within the class, and certainly on the teaching staff.

Teaching staff may be a third source of limitation, even if it is a specific feature of MetaEmotions at School program. As already described MetaEmotions at School is a train-the-trainers program and, even if all trainers (class teachers) followed the same workshops program and had the opportunity to use the same activities and guidelines available in the MetaEmotions platform, they were equally free to use them at will and to adapt them in order to meet specific school activities or specific needs of their students.

This leaves open the possibility that other factors (e.g., workload, motivation, attitudes, knowledge, stress, emotional abilities, training level) could have influenced teachers'

decision to choose an activity over another one as well as their performance. In addition, when Principals have to assign classes to a new experiment, the choice falls more often on the identification of available teachers than on the students in the classes. The referent teachers are in fact a fundamental piece in this type of research, and if it is undeniable that all the teachers involved have accomplished their best and with the utmost commitment, not all have succeeded in the same way, because of factors independent from their will, for example related to school organization. Furthermore, since intervention and control classes were intentionally selected from each school, teachers might have shared information or contents about the activities while pupils might have interacted or socialized altering the results.

Finally, sample size represents another limitation, because the overall number of participants is small and that is largely due to different size of each class.

## 5. Conclusions

In conclusion, and after considering limitations above described, the results obtained demonstrate, as expected, important changes in many students belonging to intervention classes in a very short period of time.

The results described here are not the only ones derived from this first experimentation conducted with the meta-emotion program, which has also seen its application in primary school. Given the vastness of the materials collected in the first experimentation, it seemed appropriate here to focus only on the results relating to emotional intelligence, hoping to soon have the opportunity to illustrate the other results relating to school learning, social relations between peers, as well as the interesting results relating to the satisfaction of pupils and teachers.

The applications of the MetaEmotions method are still in progress, and have already seen an important transformation, that was also suggested by this first study. The great variability of the results obtained has led us to focus on the enormous role that teachers play in socio-emotional education. As already mentioned, we are convinced that, even when you try to give a very structured curricula to teachers, asking them to follow it step by step, they always change it deeply on the base of the specific context. We simply decided to transform this "noising effect" in intervention study into a resource: teachers felt free to adapt the activities and they tried to embed emotional education in their normal curricula. Moreover, it should never be assumed that training someone to use of a method is itself a sufficient condition for that method to be applied well. To be good trainers in the field of emotions, teachers must not only possess the theoretical knowledge necessary to apply a method to be good trainers in a SEL program, but teachers must also "know how to be", and activate an imitative learning by the pupils through their behavior. They have to show them, on the one hand, to have contact with their emotional world and, on the other, to have the ability to manage emotions and not be overwhelmed by them. To this matter, a recent study [30] has shown that, in teachers, perceived emotional intelligence positively correlates with work engagement and job satisfaction, and negatively correlates with burnout. These results suggest that emotional intelligence may have a protective role in preventing negative working experiences of teachers. For these reasons, it was considered that MetaEmotions at School program could not be limited to the emotional empowerment of the students, but should also recover the emotional empowerment of the teachers themselves. For these reasons, the current applications of MetaEmotions at School program provide—as already foreseen in the MetaEmotions Test & training version of the program—the measurement of meta-emotional intelligence of teachers at ethe beginning of training. Next, teachers are returned the results obtained, in order to initiate a meta-emotional reflection on themselves, which can then facilitate them in conducting more effective emotional education for their pupils, and to pave the way for truly emotionally inclusive schools, able to prevent in school all forms of social deviance, the emergence of risky behaviors and social exclusion.

**Author Contributions:** Conceptualization, A.D.; Data curation, A.G.; Formal analysis, A.G.; Funding acquisition, A.D.; Investigation, A.D.; Methodology, A.D.; Project administration, A.D.; Resources, A.D.; Software, A.G.; Supervision, A.D.; Validation, A.D. and A.G.; Visualization, A.D. and A.G.; Writing—original draft, A.D. and A.G.; Writing—review & editing, A.D. and A.G. All authors have read and agreed to the published version of the manuscript.

**Funding:** This research was funded by the Italian Authority Warranty for Infancy and Adulthood.

**Institutional Review Board Statement:** Not applicable.

**Informed Consent Statement:** Informed consent was obtained from all subjects involved in the study.

**Data Availability Statement:** Data available on request from the corresponding author A.D.

**Acknowledgments:** This study belongs to a wider research-intervention project also involving primary school teachers and students, developed thanks to the support of the University of Palermo and not-for-profit organization MetaIntelligenze. We thank here all the teachers and students involved in the project, as well as all professionals from MetaIntelligenze that supported the different steps of the activities: Martina Enea, Teresa Guastaferro, Cinzia Gambino, Daniele Armetta, Giusy Fabiola Ferraro, Vittoria Cedro, Maria Vittoria Caiozzo, Martina Di Marco, Giusy Guella.

**Conflicts of Interest:** The authors declare no conflict of interest.

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
