# Peer review of "MetaEmotions at School: A Program for Promoting Emotional and MetaEmotional Intelligence at School; a Research-Intervention Study"

_education, doi:10.3390/educsci12090589_

Round 1
Reviewer 1 Report
Thank you for the opportunity to review this paper.
This is a quasi-experimental study of a school intervention that focuses on emotional abilities of lower secondary students
Abstract
The authors list the 5 outcomes of the train-the-trainer program, but the focus of this study is not to examine all 5 outcomes so it is misleading to include these in the Abstract. Consider removing and revising the abstract.
Feedback
Introduction
The first sentence should be re-written, it is an assumption that parents protect children from the experience of emotions, and that all children have the same home experiences. Perhaps a stronger argument can be made that school is where students interact with others outside the family (teachers, peers), and that they start to receive feedback about emotional abilities through these social interactions. Re-write the sentence that begins with “For these reasons,” on lines 28-29 to be grammatically correct, and also that the context of school provides student with the opportunity to engage with others and to learn new ways (or simply foundational) emotional skills if they haven’t already.
Consider re-wording line 48 to reflect that the present study is informed by Mayer and Salovey model, explain that model briefly (as it is a theoretical framework), then you may move onto discus s the SEL program is also informed by Mayer & Salovey. Since the work by the authors cannot be listed - is it possible to explain the MetaEmotions program briefly? Then further on, around lines 93-95 remind the readers that MetaEmotions is aligned with RULER and CASEL principles. Since the study measures emotional and metaemotional intelligence more focus on these concepts in the Introduction review is required.
Present Study
Please explain how classes were assigned a condition. Also, because this is a quasi-experimental design – it would be better to label the 2 classes conditions as intervention group and comparison group as it is isn’t a true clinical trial. Can the authors explain why a certain number of students received training alongside their teachers? Is this part of the programs’ mission? Philosophy?
Rephrase lines114-117: The study unfolded or was rolled out in steps. First, emotional and metaemotional intelligence.
Material & Methods
Participants
For lines 145-146 “As requested by principals, parent consent forms were distributed to all eligible students, and only those with parental permission participated.”
Was this study approved by any Ethics review at a university or school board level? Please indicate – as the study should have ethical clearance.
Procedure
Please indicate the actual years data collection occurred {then statement about pre-covid may be removed]
Measures
Please include validity and reliability information for all the measures created and included. Psychometric information is essential in determining the validity of these measures used. How many items per measure, and maybe even a sample question for each, and what type of scales or measures they were. Can each measure be affiliated with a specific reference or references?
The training program needs to be simplified and streamlined. Perhaps the program and different steps may be summarized in a clear, concise way in a Table.
Results
The results are presented in an easily accessible way. Please refer back to the study’s original questions and predictions.
Discussion
The discussion should speak to the results conceptually only and not provide any numerical information. More reference to literature presented in the Introduction is needed. As it stands the Discussion reads more like an extension of a Results section. What was supported and what wasn’t - what does this mean with respect to the theory any hypotheses. This section needs to be strengthened in this regard. Perhaps consider moving part of conclusion information (which reads like a discussion into the discussion section and shorten your take-home messages and be very concise with your conclusions that stem directly from your study questions and findings.
Edits and Comments:
· A thorough review for grammar and inappropriate word use is required
· Please remove the use of the word “indeed” throughout the manuscript is overused, and not required
· Add “the” before scientific community on line 31
· On line 34, remove “more and more” and replace with “increasingly studies…”
· On lines 63-68, there are some writing errors correct: remove indeed, then do not repeat “the area of” each time- simplify to using it once. Then, specifically on liune68 – the word declined may be an error - are the authors meaning to say “designed”?
· If there are 4 macro-areas why are 5 listed (lines 63-68)??
· Line 85 switch enough with aware (wrong order) line 93 the word “on” is missing after focus.
· Replace treatment with intervention/or program
· Line 402 “and” should be “an”
· On line 412 – remove the word lifetime - the authors either believe that emotional skills are malleable or not - perhaps the argument is that emotional skills is something should not be a limited program, but embedded within a curriculum from early childhood into early adulthood as life and school in general becomes increasingly more complex.
Reviewer 2 Report
Dear Authors,
I have found your paper "Meta-emotions at school" an interesting contribution to the field, covering both implementation and evaluation factors. I think the paper reads well, conveying an important message about the central role of emotional intelligence in education.
Intervention and implementation methods seem very interesting, I would congratulate you on technology use as well as adapting the program delivery to specific Italian context.
I would endorse the paper for publishing but some changes have to be incorporated first. I hope you will find those are improving your paper. I have several suggestions:
- English language and style has to be proofread additionally, there are some mistakes and some style difficulties
- In the introduction, line 41, I would suggest also stating some scientists involved in CASEL besides Daniel Goleman. He has done a lot in promoting SEL with his book but experts as Roger Weissberg contributed in a more scientific manner which I find more suitable for a paper
- When describing IE-ACCME test in lines 75 to 81 authors state that test also involves discussion of results per each participants as well as commenting results during sessions. I would recommend to authors to be careful here: 1) state clearly is the test paper-pencil or an interview and what are the down-sides if it is an interview (I would than see it as a little intervention); 2) in the procedure, it is not clear if the reflection on results part was also done with control group participants in any manner. If so, please explain in detail.
- Authors should include factor and reliability analysis of the IE-ACCME test since this is crucial for the integrity of results presented. Factor and reliability data can be part of the measures section.
- My thoughts regarding the procedure that involved 2 classes from same school, involving one in intervention group and other in control - I think that could be a possible bias source since we know that teachers talk and that even share the contents and resources. Also, kids socialize and interact. I think that could have tainted the results. Please stress that in the limitation section.
- I would expect Cohen's d accompanying ANOVA's results. Please include these in the result section.
- There is a mistake when stating CC or CG and EC or EG when differentiating groups. Be consistent here and think about calling the groups intervention groups rather than experimental
- If possible include some references why we get negative effects when evaluating interventions in the discussion. Also add similar findings reported elsewhere.
- Limitation section should also address small number of participants; the fact that teachers can choose activities independently, the fact that their stress, training level, motivation and personal emotional intelligence could be different -that all could have effected results.
- I would expect a shorter title and would recommend to authors to include that this is a pilot research.
